# Down Syndrome—Basque Alzheimer Initiative (DS-BAI): Clinic-Biological Cohort

**DOI:** 10.3390/jcm13041139

**Published:** 2024-02-17

**Authors:** Miren Altuna, Ainara Estanga, Adolfo Garrido, Jon Saldias, Marta Cañada, Maitane Echeverria, José Ángel Larrea, Patricia Ayo, Ainhoa Fiz, María Muñoz, José Santa-Inés, Valeria García-Landarte, Maite García-Sebastián

**Affiliations:** 1Fundación CITA-Alzheimer Fundazioa, 20009 Donostia, Spain; 2Debabarrena Integrated Health Organization, Osakidetza Basque Health Service, 20690 Gipuzkoa, Spain; 3Department of Medicine, Faculty of Health Sciences, University of Deusto, 48007 Bilbo, Spain; 4Donostialdea Integrated Health Organisation, Clinical Biochemistry Department, Osakidetza Basque Health Service, 20014 Donostia, Spain; 5Donostialdea Integrated Health Organisation, Radiology Department, Osakidetza Basque Health Service, 20014 Donostia, Spain; 6Atzegi, 20004 Donostia, Spain; 7Fundación Goyeneche de San Sebastián, 20018 Donostia, Spain; 8FEVAS Plena Inclusión Euskadi, 48009 Bilbo, Spain

**Keywords:** Down syndrome, intellectual disability, Alzheimer’s disease, biomarkers, aging, cognition

## Abstract

Background: Down syndrome (DS) is the most common genetically determined intellectual disability. In recent decades, it has experienced an exponential increase in life expectancy, leading to a rise in age-related diseases, including Alzheimer’s disease (AD). Specific health plans for the comprehensive care of the DS community are an unmet need, which is crucial for the early and accurate diagnosis of main medical comorbidities. We present the protocol of a newly created clinical and research cohort and its feasibility in real life. Methods: The Down Syndrome—Basque Alzheimer Initiative (DS-BAI) is a population-based, inclusive, multidisciplinary initiative for the clinical-assistance and clinical-biological research approach to aging in DS led by the CITA-Alzheimer Foundation (Donostia, Basque Country). It aims to achieve the following: (1) provide comprehensive care for adults with DS, (2) optimize access to rigorous and quality training for socio-family and healthcare references, and (3) create a valuable multimodal clinical-biological research platform. Results: During the first year, 114 adults with DS joined the initiative, with 36% of them showing symptoms indicative of AD. Furthermore, adherence to training programs for healthcare professionals and families has been high, and the willingness to collaborate in basic and translational research has been encouraging. Conclusion: Specific health plans for DS and conducting clinical and translational research on the challenges of aging, including AD, are necessary and feasible.

## 1. Introduction

Down syndrome (DS) is the most common genetic cause of intellectual disability [1,2]. In total, 96% of cases are due to the complete trisomy of chromosome 21, with only 4% attributed to mosaicism and translocations [3]. This is why DS is also referred to as trisomy 21. The incidence of individuals born with DS each year is 1 in every 700 births. The number of births in developed countries has significantly decreased over the past decades due to advances in prenatal diagnosis [4]. This is despite the increase in maternal age, a known risk factor for the development of trisomy 21. However, there has been an exponential increase in the life expectancy of individuals with DS, surpassing 60 years at present [4,5]. Therefore, the prevalence of adults with DS has not only not decreased but has increased over the last few decades.

The exponential increase in life expectancy is attributed to healthcare advancements, with particular significance placed on early corrective surgery for congenital heart defects—an undeniable milestone in the improved life expectancy of individuals with DS [6,7]. Equally indisputable are the educational and social advances that have enabled a comprehensive approach to all the needs of the DS community. This includes improved access to inclusive employment, ensuring better support for independent living at home, and the creation of adapted residential resources, among other aspects. Unfortunately, until now, the maximum effort and, consequently, the most fruitful results have been focused on the pediatric and young adult age groups. Until a few years ago, late adulthood did not receive sufficient attention. Late adulthood in the DS population occurs at an earlier age, typically starting around 35–40 years, as individuals with DS experience an aging process approximately 20 years earlier than the non-trisomic population [8].

Linked to the aging of the DS population, there has been an exponential increase in age-related diseases, with Alzheimer’s disease (AD) being unquestionably prominent among them [8,9]. DS is considered a genetically determined form of AD with virtually complete penetrance (DSAD), and it is the most common genetically determined form of AD [9,10]. The prevalence of symptomatic AD is up to 80–90% for the age of 60 years, and the mean age of symptoms of onset is of 50–55 years [8,10]. It is known that the triplication of the amyloid precursor protein (APP) gene, located on chromosome 21, is a sufficient cause for the development of DSAD [8]. Currently, DSAD represents not only the primary health challenge for the population with DS, negatively impacting their quality of life, but it is also the leading cause of death [4,11,12]. In recent years, significant progress has been made in early and accurate diagnosis of DSAD [8,10,13], but these advances have not been universally implemented due to a shortage of qualified healthcare personnel and a lack of social awareness regarding the new challenges associated with the aging of this community.

Not only is there a higher prevalence of AD in the adult population with DS, but there is also an increased risk of epilepsy [14,15] associated with AD, sleep apnea-hypopnea syndrome [16,17], sensory deficits [6,18,19], hypothyroidism [20], etc. (Figure 1). All of this demands the creation of highly specialized health plans for this community. It is also essential that such a health plan integrates with available social resources, emphasizing health promotion and prevention rather than solely focusing on early diagnosis and treatment. This strategy must include training programs targeting individuals with DS, their family members, educators, caregivers, and healthcare professionals as a key component for the success of a comprehensive health plan. Additionally, it is of vital relevance to facilitate access to research, with individuals with DS being key participants. This aims to enhance the diagnostic and therapeutic processes for various diseases affecting them and to better understand if there are variations in the physiopathology of the main age-related diseases in the DS population. The exclusion of individuals with DS from clinical research studies, especially clinical and translational trials, is another way to promote discrimination. This has systematically occurred for many years and has recently manifested itself in the non-inclusion of therapeutic trials with anti-amyloid drugs that have led to their conditional (Aducanumab) [21,22] or definitive approval (Lecanemab) [23] for the treatment of AD in non-trisomic populations.

In this context, we developed a comprehensive health and socio-healthcare plan aimed at the holistic care of the adult population with DS in our region (Basque Country) lead by CITA-Alzheimer Foundation. Our initiative was inspired by other health plans and research cohorts for DS population such as DABNI (Down Alzheimer Barcelona Neuroimaging Initiative) or those lead by the Jerome Lejeune Foundation in Paris or the London Down Syndrome Consortium (LonDownS) in the UK, but, obviously, they have been adapted to our socio-health context. The current estimated prevalence of DS is 5.6–6.7 individuals per 10,000 inhabitants worldwide [8]. Therefore, it is believed that there are potentially 400,000 individuals with DS in Europe and more than 1000 in our region of interest (Basque Country and Navarre). In this context, we provide highly specialized healthcare, close coordination with local third-sector agents (educators, caregivers, etc.), and tailored training to address the new challenges associated with the aging of this community. This training is directed at both healthcare professionals and professional and non-professional caregivers. Both the healthcare provided, training, and freely accessible dissemination on social media and online platforms are conducted in the native language of the participants. Our region is bilingual, with services available in both Basque and Spanish (in the Spanish part of the Basque Country) and Basque and French (in the French part of the Basque Country). In turn, we provide the option for individuals who wish to collaborate in clinical and translational research. This health plan and the establishment of a research cohort are named “DS-BAI: Down Syndrome—Basque Alzheimer Initiative”. Below, we present the creation of this cohort and the preliminary results obtained during its first year of establishment.

The objectives of this study are as follows: (1) to introduce the protocol for neurological–neuropsychological clinical visits and the acquisition of multimodal biomarkers in the DS-BAI cohort; (2) to provide information on the complementary training plan accompanying the DS-BAI visits; and (3) to present preliminary data from the first year of the initiative’s implementation, aiming to demonstrate its feasibility in practice.

## 2. DS-BAI: Down Syndrome—Basque Alzheimer Initiative

### 2.1. Recruitment

The recruitment is population-based, primarily involving (1) local social agents (associations and foundations dedicated to the care of the adult population with DS); (2) awareness campaigns on social media and in traditional media outlets (newspapers, television, and radio); and (3) training sessions conducted for healthcare professionals (neurology, psychiatry, and primary care services, primarily).

The inclusion criteria to join the initiative are as follows: (1) individuals with DS aged 18 years or older (regardless of (a) level of intellectual disability and (b) cognitive status, and/or the presence of active medical comorbidity) and (2) having an accessible reliable informant. The exclusion of subjects under the age of 18 is due to various reasons: (1) the healthcare organization within the territory (individuals with intellectual disabilities are often followed-up in pediatric neurology rather than adult neurology until the age of 18), (2) the tools used for neurology and neuropsychology assessments in DS differ between minors and adults (and among minors based on age ranges), (3) medical comorbidities to be addressed also differ between childhood–adolescence and adulthood, and (4) from the age of 18 onwards, it is essential for the non-incapacitated individual with DS to directly consent to their participation in the initiative, rather than relying solely on the criteria of their parents or other guardians, thereby giving a more active role to individuals with DS.

The request for a visit is made directly by the family or legal representative of the adult with DS, either by phone or through an email channel specifically activated for this purpose. At this time, information about the preferred language for receiving assistance is collected to ensure respect for linguistic rights guaranteed in healthcare. A waiting list for the initial visit is ensured to be less than one month at all times.

On the other hand, recruitment for the training program aimed at educators, caregivers, and family members is carried out through (1) informative emails and phone calls to participants in medical visits; (2) communication on social media; and (3) dissemination with the assistance of local agents involved in the care of the adult population with DS. In turn, the training for healthcare professionals is carried out through (1) direct contact with local and regional medical associations; (2) collaboration with neurology departments in local public health system hospitals; (3) engagement with primary care management; and (4) promotion on social media. All training is initially conducted in the local minority language (Basque), followed by sessions in Spanish afterward (and if there is sufficient demand, also in French). Registration for the training is open access, either through the phone or a specifically designated email address.

### 2.2. Data Collection

The data collection is carried out using forms specifically created for this purpose in RedCap [24,25], a web platform for creating databases, for both qualitative and quantitative variables, collected either during clinical visits or as a result of interpreting the results of complementary tests. Specifically, information is collected in different forms regarding the MEDICAL and NEUROPSYCHOLOGICAL VISIT: (1) personal background related to socio-educational and socio-labor development, handedness, bilingualism, place of residence, incapacitation status and degree, whether or not assistance is provided by the dependency law, etc.; (2) personal and family (first-degree relatives) pathological history; (3) list of active medications; (4) structured cognitive interview for DS population; (5) Cummings Neuropsychiatric Inventory (NPI); (6) functional autonomy scales; (7) neuropsychological evaluation including scores for each administered test both for the estimation of intellectual disability and for the estimation of cognitive status in relation to AD; (8) detailed medical history of cognitive and behavioral symptoms with identification of first symptom and date of onset, and history of other potential comorbidities; (9) registration of antropometric parameters and neurological physical examination including the administration of specific scales for the evaluation of gait disorders and possible synucleinopathy co-pathology; (10) clinical diagnosis related to cognitive status regarding AD), the presence of other medical comorbidities, and their severity; and (11) treatment and/or research participation plan and follow-up in consultations. In addition, there is another section dedicated to INTERPRETATION OF TEST RESULTS, with its specific forms: (1) blood test form providing information on extraction data and results of hemogram and biochemistry, and another form detailing the quantity of serum/plasma/whole blood and buffy coat aliquots available; (2) cerebrospinal fluid form with information on numerical results and AT(N) biomarker classification, and another form detailing the number of aliquots available; (3) electroencephalography form providing information on qualitative analysis of the recording, specifying the presence or absence of epileptiform and non-epileptiform abnormalities, and background rhythm; (4) brain magnetic resonance imaging form including the presence or absence of unexpected findings, global cortical atrophy pattern, medial temporal and parietal atrophy patterns, presence or absence of white matter lesions and their magnitude, presence or absence of microbleeds and their location; and (5) sleep study form collecting information through structured interview, sleep diary, and results of objective sleep measurement tests. Finally, there is another section dedicated to INTEREST IN RESEARCH AND TRAINING INITIATIVES with two independent forms collecting the following: (1) interest in training plan and its modality; and (2) information on interest in research, participation in visits only or in one or several of the suggested tests, and potential interest in clinical trials.

### 2.3. Ethical Issues

The international ethical recommendations for medical research in humans are rigorously followed, following the standards outlined in the Declaration of Helsinki and Spanish legislation. The necessary documentation, including the study protocol and consent model, has been approved by the Ethics Committee of Gipuzkoa and of the Basque Country. Regarding data confidentiality, compliance with the provisions of Organic Law 3/2018, of 5 December, on Data Protection and Digital Rights Guarantee, will be strictly followed. The files will be stored in an independent and encoded database.

### 2.4. Medical Visit Structure

The medical assessment is conducted by neurologists specifically trained in intellectual disability and neurodegenerative diseases causing cognitive impairment, epilepsy, and sleep disorders. They are neurologists responsible for comprehensive care, not only for neurological pathologies, and the choice of a neurology specialist is justified by a higher risk of neurological diseases, which are often more challenging to diagnose and manage. 

The first medical visit performed at the CITA-Alzheimer Foundation includes (Table 1) obtaining information about the following: (1) pathological (special attention to risk factors of developing epilepsy) and (2) non-pathological personal history (received education, languages learned, professional activity, degree of acquired autonomy, place of residence, legal incapacitation and its characteristics, among others); (3) family history (of first- and second-degree relatives); (4) current history (directed anamnesis); and (5) systemic and detailed neurological examination. The recommendations of the structured interview CAMDEX-DS (The Cambridge Examination for Mental Disorders of. Older People with Down’s Syndrome and Others with Intellectual) [26] and also Cummings’ NPI questionnaire [27] are followed. The information is also supplemented with the Lewy Body Dementia Composite Risk Score [28] to assess the probability of co-pathology with AD and Lewy body disease, a questionnaire administered to the caregivers and completed with the information from the physical examination of the participants. Additionally, the information is supplemented through self-administered functional autonomy scales by caregivers, such as the DMR (dementia questionnaire for persons with intellectual disabilities) [29]. 

The physical examination is conducted in a structured manner, starting with vital signs assessment, determination of body mass index, cardiopulmonary auscultation, followed by a systematic neurological examination (personal, temporal, and spatial orientations; cranial nerves; language; muscle strength and sensory evaluation; balance assessment). Additionally, protocols for assessing synucleinopathy comorbidity (motor subscale of SCOPA) [30] and gait and balance (Tinetti) [31] are applied.

If feasible (not applicable in cases of profound intellectual disabilities and/or advanced dementia), while gathering information from family/caregivers during the medical history, the participant undergoes a systematic neuropsychological assessment. 

The estimated duration of an initial joint neurological–neuropsychological assessment between neurology and neuropsychology is 90 min, and 60 min for the structured annual follow-up (reduced time due to less emphasis on collecting both pathological and non-pathological histories) (Figure 2). At the end of the medical visit, individuals are informed about the option to participate in research by providing anonymized clinical data +/− undergoing potential complementary tests. It is clearly explained whether the purpose of conducting these additional tests is for healthcare (direct application in clinical practice) and research or purely for research purposes. The scheduling of the agreed-upon tests then takes place, and participation in these tests is entirely voluntary. Participants may choose to undergo all or some of the tests. Non-acceptance of participation in research (in its entirety or in part) does not affect access to provided healthcare services.

### 2.5. Neuropsychological Assessment

The initial neuropsychological assessment performed in CITA-Alzheimer Foundation includes (Table 1) estimating the intelligence quotient through (1) the Kaufman Brief Intelligence Test Second Edition (K-BIT2) [32] and (2) based on information obtained from the medical history conducted through neurology (including information from functional autonomy questionnaires). Subsequently, the current cognitive status is assessed using neuropsychological batteries adapted for intellectual disabilities [33] to estimate overall cognition and specific domains [34], with particular attention to episodic memory [35]. 

The neuropsychological assessment will not be applicable in all cases of profound intellectual disability and half of the cases of severe intellectual disability, regardless of cognitive status. It will also not be applicable in cases of advanced dementia, regardless of the level of premorbid intellectual disability.

The duration of the neuropsychological evaluation during follow-up is shortened by 30 min as there is no need to conduct a new estimation of the intelligence quotient (K-BIT2).

A comprehensive neuropsychological assessment will be conducted annually for all participants identified as cognitively healthy based on a consensus diagnosis among neurology and neuropsychology professionals. Those suspected of possible cognitive decline in addition to intellectual disability, regardless of the cause of such deterioration, will undergo assessments every six months. If more frequent visits by neurology are required, neuropsychological evaluations will not be conducted within intervals shorter than 6 months due to their limited diagnostic and monitoring value and to mitigate potential learning effects.

### 2.6. Blood Sample Collection and Analysis

All participants in the initiative are recommended by the medical team to undergo at least annual blood sample extraction. Fasting is not required for the blood sample extraction; however, detailed information about the participant’s last intake is collected. This allows for the extraction during the medical visit itself, enhances recruitment capacity, and minimizes participant discomfort. This extraction is voluntary, and no participant is excluded from the clinical cohort for refusing to provide the sample. In the obtained blood sample, initially, biochemical parameters with potential clinical relevance are determined (Table 2). Certainly, this analytical determination protocol could vary based on each participant’s personal history. Additionally, depending on the results obtained, the frequency of analytical controls may be adjusted, transitioning from annual to semiannual, always guided by practical clinical criteria.

Moreover, all participants agreeing to blood sample extraction for clinical purposes are offered the option to provide an additional sample for storage and potential use in future research studies (2 tubes EDTA 5 mL, 2 serum tubes 5 mL, 1 whole blood tube, and Buffy coats). Afterwards, it is processed (centrifugation 3500 rpm at 4 °C for 15 min) to obtain the buffy coats, serum, and plasma and stored at −80 °C. Refusing research purpose does not limit the collection of samples for clinical purposes, including their subsequent analysis and communication of results. All of this is conducted in accordance with the standards defined by the local ethics committees.

### 2.7. Brain Magnetic Resonance Imaging

At the conclusion of the baseline clinical visit, all participants are offered the option to enhance the study, in addition to blood sample extraction, by undergoing structural brain magnetic resonance imaging (MRI). The purpose of the MRI is specified during the visit, whether it is for mixed clinical (e.g., in cases of additional cognitive impairment, de novo neurological focal signs, or a history of epileptic seizures) and research purposes or purely for research purposes. The MRI is conducted without intravenous contrast administration and only after confirming the absence of contraindications. 

The MRI is conducted within the first three months of the neurological–neuropsychological evaluation. Its acquisition is under resting conditions on a SIEMENS 3T Magnetom Trio Tim scanner (Siemens, Erlangen, Germany) in combination with a 32-channel head receiver located at the CITA-Alzheimer Foundation. Acquisitions covering the whole head are performed, and the AutoAlign tool (Siemens) is used to increase homogeneity between acquisitions of the same subject (longitudinal) or between acquisitions of different subjects. The protocol to be performed includes the next sequences: T1 (3D), FLAIR (3D), T2 (3D), DWI, and SWI. The total duration of the protocol is 33 min. Sedation is not administered for its execution under any circumstances within this initiative.

A clinical interpretation of the findings is performed by neurologists, with support from neuroradiologists. The presence or absence of diffuse cortical atrophy and its degree are determined, along with cortico-regional atrophy, especially in the medial temporal region (Scheltens’ ATM scale) and the parietal region. The report also provides structured information on the presence or absence of cerebral vascular pathology and white matter lesions, indicating their severity and location (Fazekas scale). Additionally, a systematic review is conducted for the presence or absence of microbleeds, including their location and number. Furthermore, the images are stored for future research studies.

The consideration is made for performing longitudinal magnetic resonance imaging at the 2-year mark from the baseline, in all those with good tolerance to the baseline MRI, complete clinical follow-up (with or without additional complementary studies), regardless of changes in cognitive status, and regardless of the development of neurological complications (seizures or de novo neurological focal signs).

Refusal of both the baseline and longitudinal MRI does not alter the clinical approach to the participant in the initiative, as the participation is entirely voluntary.

### 2.8. Cerebrospinal Fluid Sample 

Following neurological–neuropsychological evaluation, in participants with suspected cognitive decline possibly related to neurodegenerative causes, especially Alzheimer’s disease (AD), the option of undergoing a lumbar puncture for biomarker analysis of amyloid (A), tau (T), and neurodegeneration (N) is proposed. Before its execution, the absence of absolute contraindications is confirmed. A blood test, including a complete blood count and coagulation profile, is conducted in the previous week. Additionally, previously, a physical examination, including a fundoscopic examination to rule out signs of increased intracranial pressure, and/or a structural neuroimaging test are performed. 

The lumbar puncture is performed in a sitting position, following the administration of a local anesthetic with a subcutaneous needle, in the L3-L4 or L4-L5 lumbar space. Extraction is performed with a 25G thick beveled needle with the corresponding aseptic measures and ensuring a post-puncture rest of at least 30 min. A total of 1 mL is extracted for routine clinical biochemical analyses (cell count, proteins, and glucose); 2.5 mL for AT(N) biomarkers (Aβ1-42, Aβ1-40, t-tau, p-tau); and for those who consent, 15 mL is extracted for storage at −80 °C. The surplus CSF samples, of significant biological value and obtained with difficulty due to initial rejection because of the invasive nature of the technique, are decided to be stored for potential future studies. These studies could aim to identify new diagnostic and prognostic biomarkers related to aging in DS and specifically with AD, enhancing understanding of the pathogenesis of AD and other associated comorbidities, and potentially identifying new therapeutic targets. It has been demonstrated that the amount extracted is safe, without increasing the risk of post-puncture headache, specifically in the DS population [36]. 

For those without a clear clinical indication (no suspicion of cognitive decline in addition to intellectual disability), the option of cerebrospinal fluid extraction will also be presented for research purposes after the baseline visit, with complete voluntariness. In both those with purely clinical, mixed clinical and research, and/or purely research purposes, an attempt will be made to obtain a blood sample (following the explanation in the blood samples section) immediately after the lumbar puncture. Similar to the brain MRI, the consideration of a longitudinal lumbar puncture, two years after the initial one, will be proposed for all subjects with a lumbar puncture and without complications (primarily without post-puncture headache) after the baseline visit. However, refusing to undergo these procedures will not result in any changes to the clinical care provided.

### 2.9. Video-Electroencephalogram

After the completion of the baseline clinical visit, all participants are going to be proposed to undergo a prolonged 60 min video-electroencephalogram (Video-EEG) with at least 32-channel systems for video-EEG recording, preferably with sleep deprivation. Electrodes are strategically placed on the scalp according to the International 10–20 system. Additional electrodes are applied for specific purposes, such as for capturing muscle activity (EMG) and eye movements (EOG). Simultaneous video recording is performed, and during the recording session, trained technicians or clinicians monitor the patient’s behavior and EEG activity in real-time. 

The extended duration of the recording and sleep deprivation may enhance diagnostic performance for detecting both epileptiform and non-epileptiform abnormalities. Intermittent light stimulation and hyperventilation are used as activating maneuvers. The hyperventilation, as long as the patient cooperates, will last at least 3 min. The light stimulation, estimated to last 5 to 6 min, adheres to the following specifications: the lamp is positioned 30 cm from the patient; the response is assessed with eyes open and, after 5 s, with eyes closed; a frequency sequence of 1, 2, 4, 6, 8, 10, 12, 14, 16, 18, 20, 60, 50, 40, 30, 25 Hz is used; flashes are delivered in 10 s trains per frequency with a minimum separation of 7 s (initially ascending frequencies from 1 to 20 Hz and then descending from 60 to 25 Hz). Information is collected about the following: (1) qualitative (presence or absence of epileptiform and non-epileptiform abnormalities and their characteristics) and (2) quantitative assessment (such as the background activity frequency) of the test.

The indication for performing video-EEG is particularly relevant in those with a diagnosis of de novo epilepsy, regardless of cognitive status (symptomatic or asymptomatic for Alzheimer’s disease). In such cases, if there is intolerance, and given its clear clinical practice purpose, the consideration of a 30 min video-EEG (instead of 60 min) without sleep deprivation is proposed. Routine video-EEG determination may also be considered for those with symptomatic AD who are unlikely to tolerate prolonged recordings with sleep deprivation. In such cases, there may be a greater potential benefit for the detection of subclinical epileptiform activity.

For those who agree to undergo video-EEG and in the absence of semiological data suggesting the presence of de novo epileptic seizures or significant changes in their semiological characteristics and/or an increase in frequency (for those with known epilepsy), the proposal for annual video-EEG will be made. However, in the event of the onset of episodes compatible with epileptic seizures or changes in their characteristics, a video-EEG will be conducted as close as possible to the episode to enhance diagnostic yield. As with the rest of the complementary tests, the refusal of both baseline and longitudinal video-EEG will not result in any reduction in the quality or quantity of specialized healthcare received. This is also true in cases where acceptance is limited to clinical practice, without a research objective.

### 2.10. Sleep Studies

All participants in the medical visit undergo a focused medical history to gather information on the characteristics of sleep quality and quantity, specifically using Oviedo Sleep Questionnaire [37,38] and Epworth Sleepiness Scale [39,40]. They are informed about the increased risk of developing obstructive sleep apnea (OSA) within the population and the recommendation for a screening study for OSA in all adults with DS, even in the absence of supportive semiological data. For those who understand and accept the relevance of this study, coordination will be carried out with the primary care medical service for the implementation of ambulatory respiratory polygraphy. The sleep study is currently not conducted by the DS-BAI research team; instead, it is carried out by the primary care physician in coordination with the DS-BAI initiative’s medical professionals. Due to limitations in accessing polysomnography in our territory, and given that OSA is the most prevalent sleep disorder in the population, respiratory polygraphy (collects information on at least airflow, thoracoabdominal movements, and oxygen saturation) is employed. This test is conducted overnight at the patient’s home and may be repeated for a second night in the case of an incomplete registration and/or incongruent data. During the week including the night of respiratory polygraphy, sleep quality and quantity are documented in a sleep diary. The results of the respiratory polygraphy (including apnea–hipopnea index or IAH) are communicated to the DS-BAI research team, who then assess the appropriateness of specific treatment based on each participant’s circumstances. In the event of a detection of moderate to severe obstructive sleep apnea (OSA), treatment with continuous positive airway pressure (CPAP) will be considered. CPAP therapy will be specifically prescribed by the sleep unit professionals at each reference hospital in close collaboration with the primary care physician of each participant. Information on CPAP tolerance and adherence is collected during follow-up medical visits at DS-BAI, and in case of difficulties, direct contact is made with the Sleep Unit. If anomalies other than OSA are detected through medical history, sleep diaries, and/or specially designed questionnaires, the neurologist within the DS-BAI initiative will make the appropriate adjustments to treatment (pharmacological and non-pharmacological), with monthly or quarterly telephone follow-ups (depending on the severity and nature of the disorder) until resolution or stabilization. Looking ahead, to optimize the diagnostic process for OSA and other sleep anomalies, the incorporation of home-use devices such as WatchPAT is proposed [41,42]. These devices allow for a simultaneous acquisition of actigraphic data (sleep latency, number of awakenings, total sleep time, sleep efficiency) and cardiorespiratory parameters. They have already been approved by the Food and Drug Administration (FDA) and have been specifically studied in the population with DS [43]. 

In summary, the proposal of the DS-BAI initiative is an active screening for OSA in all participants. Currently, there is no option available, nor is there any foreseeable availability, for conducting polysomnography on all participants. Therefore, diagnosis is based on structured anamnesis and respiratory polygraphy, with the idea of incorporating devices such as WatchPAT or similar ones that could be directly applied by the DS-BAI research team to facilitate the longitudinal monitoring of participants at least biannually. Currently, repeat respiratory polygraphy is planned if there is clinical suspicion based on anamnesis and/or initial respiratory polygraphy on the borderline of normality regarding the AHI.

### 2.11. Creation of Research Platform 

In the context of the DS-BAI, an internal committee to evaluate research projects requesting access to clinical evaluation data, CSF and/or plasma samples, video-electroencephalography tracings, and/or brain magnetic resonance imaging from subjects has been created. The idea is to facilitate access to samples for all members of the Basque Network of Science and Technology and collaborating entities such as Navarrabiomed. However, this will be performed by establishing a specific collaboration agreement for each project and providing specialized support in intellectual disability and neurodegenerative diseases such as AD to potential collaborating entities, not limited to those located in the Basque Country.

### 2.12. Training and Dissemination of Knowledge

In addition to medical–neuropsychological visits, and the other complementary test performance, the DS-BAI initiative includes the dissemination of knowledge to society as a whole, educators, professional and non-professional caregivers of adults with DS, and healthcare professionals. All of this is provided freely (at no additional cost), in the native language of the target audience (Basque, Spanish, and, if necessary, French), and in a mixed format of in-person (in CITA-Alzheimer Foundation, Donostia, Basque Country) and online sessions (to address both the digital divide and ensure maximum information dissemination). Information is provided in both written and audiovisual formats.

*Training for healthcare professionals* is carried out in collaboration with local medical departments and professional organizations. Presence is ensured in each province of the territory, on different dates, to reach a maximum audience. Furthermore, the program’s quality and rigor are guaranteed, ensuring that all educational activities are accredited by the relevant scientific–technical authorities for healthcare professionals.*Training sessions for educators and professional/non-professional caregivers* are conducted in structured sessions on a monthly basis, with flexible topics based on the interests/concerns expressed by the participants. Subsequently, open access videos are created and shared on the internet to ensure the dissemination of knowledge. This serves not only as a platform to expand knowledge for educators and caregivers but also to bridge the gap between highly specialized healthcare professionals and caregivers. It helps identify unmet needs within the community that go beyond the healthcare realm.

Simultaneously, the multidisciplinary team that comprises the DS-BAI initiative is responsible for disseminating the new reality arising from the aging of the population with DS to society at large. This is achieved through regular publications on social media and collaboration with traditional media outlets. It is believed that spreading this knowledge is crucial to ensuring the inclusion of the DS community in society throughout their adult lives, including the symptomatic phases of AD. Additionally, this outreach effort may garner increased social support to advocate for the overall well-being of the community across their lifespan.

## 3. First-year Experience of DS-BAI Initiative

### 3.1. Clinical and Research Benefits

Recruitment for the DS-BAI initiative commenced on 1 January 2023, following the approval of the Gipuzkoa Committee (a) and the Ethics Committee of Euskadi (b). Committee (a) authorized the initiation of the health plan, which includes medical and neuropsychological visits for the adult DS population, as well as the performance of complementary tests but only for direct healthcare purposes. Subsequently, Committee (b) expanded the objectives of the cohort, allowing for the establishment of the clinical-biological cohort with the aim of creating a unique research platform that places individuals with DS at the center of pioneering, clinical, and translational research. 

In the first year of the initiative, more than the estimated 15% of the entire population (pediatric and adult) with DS in the territory of the Basque Country was visited as part of the initiative. In total, 114 participants, mostly men (53.8%), with an average age of 46.3 years (ranging from 18 to 65 years) and with varying levels of intellectual disability (40.4% mild, 49.5% moderate, 9.2% severe, and 0.9% profound) joined the initiative (Table 3). 

The level of acceptance of the initiative was very positive, with only one participant who did not agree to the transfer of anonymized clinical data for research purposes after the first visit and chose not to be part of the initiative. Other two users did not accept in-person clinical follow-up after agreeing to the use of clinical data for research purposes, but they accepted remote monitoring as long as there were no significant changes in their clinical condition. On the other hand, all those who agreed to the blood test for healthcare purposes also consented to the use of surplus samples for research purposes, constituting more than two-thirds of the total sample. The proposal for MRI and lumbar puncture was also well received, with more than 50% agreeing to undergo MRI and 34.2% opting for lumbar puncture.

It is noteworthy that among those who joined the initiative, 36% of the sample showed symptoms of AD, meaning they had cognitive impairment in addition to the intellectual disability associated with DS. The most remarkable aspect is that only 5.7% had a prior diagnosis before the visit, and none had received specific symptomatic treatment applicable to DSAD. This alone justifies the need and the healthcare benefit of implementing the current initiative. The average age of those diagnosed with DSAD in the prodromal stage in our sample is 54.1 years (ranging from 42 to 62 years), and for established or dementia DSAD, it is 54.3 years (ranging from 42 to 65 years). As expected, the prevalence of epilepsy increases in symptomatic stages in our sample: 4.7% in those asymptomatic for AD versus 30.8% in those symptomatic for DSAD. 

While the focus of the current initiative is evidently on neurological pathology, it is not limited to that, as participants benefit from a comprehensive care plan specifically tailored to DS. There is a high prevalence of the following medical comorbidities: hypothyroidism up to 56.5% in females compared to 38.9% in males; affective and/or behavioral symptoms with interference in quality of life and/or functionality not related to neurodegenerative processes in up to 28.3%; cataracts with interference in visual acuity up to 35.2%; neurosensorial hearing loss with an indication for specific treatment in up to 13%, and symptomatic hyperuricemia in up to 7.9% of the sample, among others. Many of these diagnoses and subsequent medication adjustments are made in the context of the current initiative. At the same time, the initiative has allowed for the reduction of medication with clinically questionable indications, especially in a population with excessive psychotropic polypharmacy in the first visit (4% on chronic benzodiazepine treatment; and 16% of the sample on antipsychotic treatment, among which 6.3% are on haloperidol and 37.5% are on risperidone), with not-insignificant extrapyramidal and cognitive side effects.

In addition to the already mentioned clinical and healthcare benefits in the first year of the initiative, the creation of a platform for clinical-biological data exploitation has sparked interest from various local research groups to promote the identification of the following: (1) new biochemical and epigenetic biomarkers for AD, epilepsy associated with AD, and other medical comorbidities associated with aging; (2) identification of modifiable and non-modifiable risk factors associated with the phenomenon of early aging and increased risk of developing symptomatic AD; (3) development of new pharmacological and non-pharmacological treatments for AD and associated comorbidities; and (4) development and implementation of more accessible tools for the diagnosis and monitoring of epilepsy and sleep disorders, among others. In other words, this initiative has created bridges between local research groups with a clinical, translational, and basic research focus, placing, for the first time, in the central axis of the proposal in our territory (the Basque Country), the group of adults with DS and their closest environment. We hope to present the results obtained from these collaborations in the near future, while also expanding the network of collaborations at the national and international levels. 

### 3.2. Practical Application of the Training Plan

The implementation of the training plan for healthcare professionals has demonstrated the need for and interest in receiving quality health information in the professionals’ native language. In fact, it has been surprising to see a better response (number of participants) in the local language, considered a minority (Basque). Therefore, greater and better health training should always be inclusive regarding the linguistic rights of professionals. This proposal, in addition to being equitable in terms of linguistic rights, has allowed for progress toward specialized healthcare for the DS community and greater integration/collaboration between primary care and highly specialized units, as well as with local third-sector agents. This increases the possibility of the early diagnosis of age-related diseases and access to potential modifying and/or symptomatic treatment, and it reduces the risk of health discrimination against the community of DS. Aspects to be confirmed in the longitudinal follow-up of the initiative.

In addition, the DS-BAI initiative promotes collaboration among healthcare professionals, educators, and family members, providing information tailored to the specific needs of each setting. This contributes to enhancing the understanding and management of health in the DS community, promoting population-level health promotion and prevention strategies. The active participation of family members, caregivers, and educators in training sessions facilitates the creation of educational materials adapted to the needs of each environment, taking into account the differences between healthcare professionals and other contexts such as work, academic, and family settings.

Finally, and not less important, awareness within society about the need for active inclusive policies has been significantly improved, not only being directed towards the pediatric and/or young adult population with Down syndrome. There has been greater engagement on social media and in the media regarding news related to DS since the beginning of the initiative.

### 3.3. Limitations

Our current work aimed to present a protocol and the first-year experience regarding its practical applicability, rather than conducting an exhaustive characterization of the recruited sample, as recruitment is ongoing and there are limited longitudinal follow-up data available for the participants.

Our initiative involves establishing a cohort with dual clinical and research purposes, which presents various challenges. Despite our efforts to recruit a population-based sample, collaboration limitations and an underrepresentation of certain age groups may compromise the cohort’s representativeness. Additionally, the absence of a diagnostic confirmation through a karyotype analysis affects the certainty of our sample’s composition, with an unknown number of subjects with complete trisomy, mosaicism, or translocations.

Furthermore, the lack of mandatory biomarker studies on Alzheimer’s disease undermines diagnostic precision. However, we aim to mitigate this through protocol studies and follow-up procedures. Although our cohort does not include PET-FDG, PET-amyloid, or PET-tau studies, we compensate by including a large number of individuals with AT(N) biomarker studies on CSF and cerebral MRI studies.

The inability to access polysomnography may limit our diagnostic capacity and characterization of sleep within the sample. To address this, we collect sleep diaries and questionnaires on sleep quality and quantity, and we conduct home respiratory polygraphy. In the future, we hope to utilize FDA-approved home devices, like WatchPAT, for screening sleep apnea syndrome, which offer additional information beyond cardiorespiratory parameters.

Our primary objective is to demonstrate the feasibility of our mixed clinical and research cohort and develop specific training plans to support its implementation. The anticipated increase in sample size, follow-up visits, and longitudinal sample collection is crucial for our success.

It is worth noting that our study reflects the racial composition of the Basque Country, with relatively low representation of other races/ethnicities different from White Caucasian, which may affect the generalizability of our findings. Finally, we acknowledge the importance of addressing linguistic rights and exploring the impact of bilingualism, recognizing both challenges and opportunities for future research. To date, due to the limited sample size, we have been unable to explore the impact of conducting neuropsychological assessments in a language different from the native language (due to the absence of validated tests in Basque) and the potential benefits of bilingualism on the cognition of adults with DS.

## 4. Discussion

The commitment to care for the late-adult population with DS (>35–40 years) is relatively recent, and comprehensive healthcare models for individuals with DS in this age range, both in our environment and internationally, are limited [8,44]. The few that exist have demonstrated that the establishment of highly specialized care units, staffed with professionals specifically trained in intellectual disability and cognitive and behavioral pathologies [45], and with access to diagnostic biomarkers for AD [10], has a direct impact on the overall health of adults with DS and also on the quality of life of their caregivers. The benefit obviously extends beyond the optimal management of AD and associated comorbidities, enabling the creation of unique models for the comprehensive care of the DS population.

However, the benefits of establishing a cohort of late-adult population with DS go beyond mere healthcare. It allows for placing individuals with DS at the center of innovative research projects [8], using multimodal biomarkers for both diagnosis and prognosis, evaluating potential new pharmacological treatments with modifying effects on the clinical-biological course of AD, among others. Excluding individuals with DS from research projects is another way of marginalizing them, a practice that has, unfortunately, occurred over the years. Therefore, achieving an optimal clinical-biological phenotyping of late-adults with DS should be a priority. The benefit of comprehensive clinical phenotyping, while primarily focused on AD as the primary comorbidity and the leading cause within the population, also applies to other common medical comorbidities, particularly those with potentially effective modifying and/or symptomatic treatments.

Furthermore, the DS population can help us better understand the complex pathogenesis of AD, beyond the deposition of amyloid and tau [3]. Specifically, multiple genes linked to the immune system are found on chromosome 21, and it is well known that neuroinflammation is a phenomenon associated with AD. Additionally, specific abnormalities in the epigenetic mechanisms responsible for regulating the expression of genes involved in the immune response have been described in DSAD [46]. The dysregulation of cellular functions such as autophagy, lysosomal activity, and mitochondrial function has also been described from early preclinical stages of DSAD. These mechanisms are also linked to other forms of AD, and, therefore, the knowledge gained in DSAD could be applicable to the general population [47]. The fact that it is feasible to provide longitudinal monitoring to the entire adult population with DS in a specific territory makes this population especially attractive to learn more about AD pathology. This monitoring allows for characterizing, clinically and biologically, all sequential changes that occur in the adult DS population, a genetically determined form of AD. It provides a unique opportunity to identify clinical and biological changes that occur in preclinical stages of AD (related to amyloid deposition +/− tau in the absence of symptoms) and factors that influence progression to a symptomatic stage of AD and, within this, to a stage of functional autonomy loss. It has been described that the biological course of sequential changes in AT(N) biomarkers in the CSF of individuals within the continuum of AD does not differ significantly in DSAD from autosomal dominant and sporadic forms of AD, and similar results have been obtained in plasma [48,49]. The predictable age of symptom onset, with much less variability than in sporadic forms, and a shorter duration of the symptomatic phase of AD also facilitate monitoring [3,44].

The protection against atherosclerotic-caused cerebrovascular damage, not linked to amyloid angiopathy, in the DS population also makes them a population of special interest. This allows for studying the impact of AD without the influence of vascular pathology throughout the entire AD continuum, something very rare in sporadic AD forms [50,51,52].

For all these reasons, they represent the optimal population, provided they undergo comprehensive clinical-biological phenotyping, to test new pharmacological treatments, including their potential application in preclinical stages [53], identify new diagnostic and/or prognostic biomarkers, study interactions between AD and other neurological comorbidities (with special attention to epilepsy [14,54] and sleep disorders [16]), and, of course, advance our understanding of the complex pathogenesis of AD. In fact, DSAD has been one of the factors that has kept the amyloid hypothesis alive in the pre-Donanemab, Lecanemab, and Aducanumab era, despite multiple previous failures of anti-amyloid drugs. Recently, specific international initiatives have been established to facilitate access to pharmacological clinical trials for individuals with Down syndrome (DS), especially in preclinical and early symptomatic stages of Alzheimer’s disease (AD). Among these initiatives, The Alzheimer’s Clinical Trial Consortium—Down syndrome (ACTC-DS) stands out, with the Trial Ready Cohort—Down Syndrome (TRC-DS) being the first global initiative underway [55]. The TRC-DS aims to demonstrate the feasibility of including individuals with preclinical and early symptomatic DS-associated Alzheimer’s disease (DSAD) in pharmacological clinical trials and to optimize the capacity of monitorize the clinical and biological progressions of DSAD. Furthermore, looking ahead, the population with DS may constitute the ideal population to better understand the pathophysiology of epilepsy associated with AD, given that the risk of epilepsy in the context of DSAD is higher than in other forms of DSAD [14]. They would be the ideal population to test anti-seizure medications as potential modifiers of the course of AD by reducing neuronal hyperexcitability [56]. Additionally, they would evidently help to better characterize neuronal hyperexcitability across the continuum of AD and identify potential biomarkers linking such hyperexcitability to synaptic damage such as axonal damage [57]. On the other hand, they are the ideal population to study the effect of OSA on the development of AD in the absence of other cardiovascular risk factors such as arterial hypertension. Hence, they are the ideal population to investigate OSA as an independent risk factor for AD development, its effect on core AD biomarkers [16,58,59], and the potential reversibility of damage with CPAP treatment.

## 5. Conclusions

Providing comprehensive care for the population of adults with DS, who constitute the most common genetically determined cause of AD with virtually complete penetrance (DSAD), is currently an unmet need. Creating specific plans to address the needs of this population is a duty, and offering them an active role in pioneering research on aging-related challenges, including AD, is a priority. Researching with adults with DS is the best way to create an inclusive population, and, in addition to directly benefiting them and their relatives and caregivers, it can have a positive impact on the health and scientific knowledge applicable to society as a whole.

## Figures and Tables

**Figure 1 jcm-13-01139-f001:**
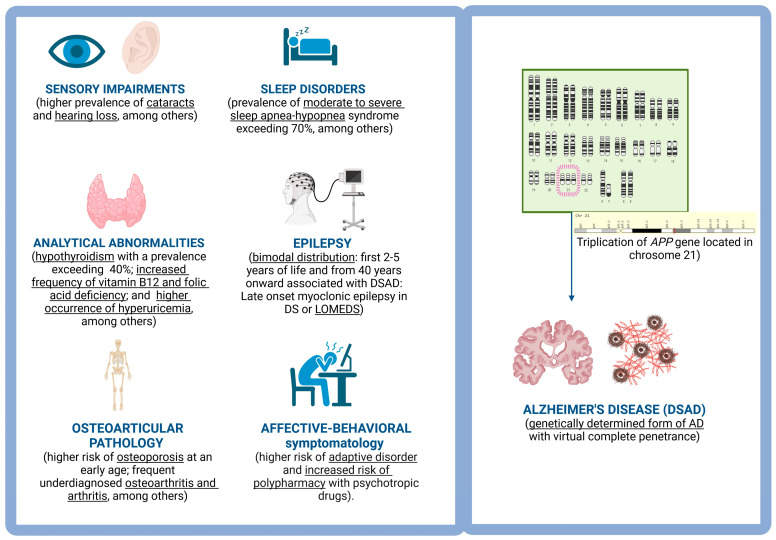
Most frequent comorbidities related to aging in DS. Figure created by Biorender.com.

**Figure 2 jcm-13-01139-f002:**
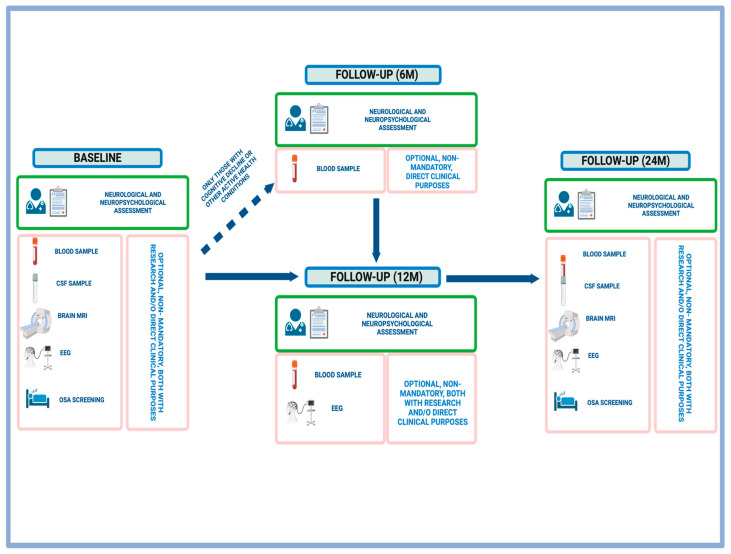
Structure of Down Syndrome—Basque Alzheimer Initiative (DS-BAI) cohort visits. Created using Biorender.com. CSF: cerebrospinal fluid; MRI: magnetic resonance imaging; EEG: electroencephalogram; OSA: obstructive sleep apnea.

**Table 1 jcm-13-01139-t001:** Summary of the structure of the baseline medical (neurological) and neuropsychological assessments of all participants of the DS-BAI cohort.

MEDICAL VISIT	NEUROPSYCHOLOGICAL ASSESSMENT
**Self-administered functionality scales** (DMR). **Non-pathological personal history** (life history, degree of autonomy, socio-educational and work trajectories, day-residential social resources, information about the residence, legal guardianship). **Pathological personal history** (systematic collection of medical-surgical history with start and end dates and the established treatment). **Family history** (collection of neurological-psychiatric or other relevant history if considered). **Current history** (structured interview following CAMDEX-DX recommendations, Cummings Neuropsychiatric Symptom Inventory, and Lewy Body Composite Dementia Risk Score). **Physical examination** (systemic physical examination, determination of arterial pressure, heart rate, and body mass index; detailed neurological examination including Tinetti scales for gait and balance, and the motor subscale SCOPA).	**Kaufman Brief Intelligence Test Version 2 (K-BIT2).** **Comprehensive Neuropsychological Assessment Battery**: CAMCOG-DS. **Episodic Memory Test** (Modified Cue Recall Test—mCRT). **Working Memory/Attention Assessment** (Direct and Indirect Digit Span; Image Cancellation Test). **Barcelona Test for Praxis Evaluation.** **Cats and Dogs Test.** **Verbal Fluency** (Animals). **Abstract Thinking** (Barcelona Test).

DMR: Dementia Questionnaire for Persons with Mental Retardation (DMR); CAMDEX-DS: The Cambridge Examination for Mental Disorders of Older People with Down’s Syndrome and Others with Intellectual Disabilities; SCOPA motor: Scales for Outcomes in Parkinson’s Disease—Motor Function; K-BIT2: Kauffman Brief Intelligence Test Version 2; CAMCOG-DS: cognitive and self-contained part of the Cambridge Examination for Mental Disorders of the Elderly with Down Syndrome and Others Intellectual Disabilities; mCRT: modified Cued Recall Test.

**Table 2 jcm-13-01139-t002:** Summary of blood test protocol for clinical purposes.

BLOOD SAMPLE: CLINICAL PRACTICE
**Complete blood count.** **Liver function and cholestatic enzymes** (AST (GOT), ALT (GPT), GGT). **Renal function and renal function and electrolytes** (creatinine, urea, sodium, potassium), **Lipidic profile** (total cholesterol, triglycerides, HDL cholesterol, LDL cholesterol). **Thyroid function and antithyroid antibodies** (TSH, Free T4, TPO). **Uric acid** **.** **Glucose and HbA1c** **.** **Calcium metabolism** (Intact PTH (PTHi), 25-OH Vitamin D).

GOT (glutamyl oxaloacetic transaminase); GPT (glutamyl pyruvic transaminase); AST (aspartate aminotransferase); ALT (alanine aminotransferase); HDL (high-density lipoprotein); LDL (low-density lipoprotein); TSH (thyroid-stimulating hormone); T4 (thyroxine); TPO (thyroid peroxidase antibody); HbA1c (glycosylated hemoglobin or hemoglobin A1c); PTH (parathyroid hormone).

**Table 3 jcm-13-01139-t003:** Main demographic and clinical characteristics at the first visit of the participants of the DS-BAI.

		Asymptomatic DS N = 72	Prodromal DSAD N = 11	Dementia DSAD N = 31	
Female (%)		31.5%	70%	13.2%	*p* = 0.274
Age (mean +/− SD)		42.4 +/− 8.1 years	54.1 +/− 5.4	54.3 +/− 5.5	*p* < 0.001 *
Intellectual disability	mild	54.2%	18.2%	16.1%	*p* = 0.002 *
	moderate	37.5%	81.8%	64.5%	
	severe/profound	8.3%	0%	19.4%	
Epilepsy (%)		4.7%	9.1%	41.9%	*p* = 0.001 *
Affective or behavioral symptoms (%)		23.6%	45.5%	58.1%	*p* = 0.436
Psychotropic medication (%)	Antidepressants	2.8%	27.3%	23.1%	*p* = 0.082
	Benzodiazepines	1.4%	0%	11.5%	*p* = 0.072
	Antipsychotics	17.2%	9.1%	23.1%	*p* = 0.632
	Anti seizure medications	9.4%	0%	34.7%	*p* = 0.010 *
Hypothyroidism (%)		31.5%	33.1%	57.7%	*p* = 0.436
Sensory deficits (%)	Visual deficits	28.1%	18.2%	53.8%	*p* = 0.040 *
	Hearing deficits	8.8%	9.1%	23.1%	*p* = 0.127

* *p*-value ≤ 0.05. Indicates a statistically significant result.

## Data Availability

Data are contained within the article.

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
