# Peer review of "Down Syndrome—Basque Alzheimer Initiative (DS-BAI): Clinic-Biological Cohort"

_jcm, 2024, doi:10.3390/jcm13041139_

Round 1

Reviewer 1 Report

Comments and Suggestions for Authors

Line 231 - what is "white series"? I am not familiar with what this is referring to. Is there another name for it?

Can you also comment on what you think some of the limitations of your program / study are? 

Overall, fantastic work in setting up a model program that is not only inclusive of adults with developmental disability (as you say, an often ignored and marginalized population) in both clinical and research aspects, but also having such robust results, knowledge translation, and societal impact in such a short timeframe. I only wish there were more programs like this in different places and for different conditions affecting adults with various developmental disabilities. 

Comments on the Quality of English Language

There are some odd word choices (e.g., "hetero administration" - line 162) and some minor grammatical issues throughout the paper that would benefit from review and editing from someone whose first language is English. Also, avoid the use of contractions such as " That's... " in a formal paper -- write it out as " That is... ". 

Author Response

Thank you very much for your suggestions.

We have corrected the term "White Series" to "Buffy Coat," which is the correct term.

Additionally, following your suggestions, we have incorporated a limitations section and have made minor changes to the wording of the text as per your recommendations.

Reviewer 2 Report

Comments and Suggestions for Authors

JCM-Manuscript ID-2848414

Reviewer Report

I appreciate the opportunity to review your paper, and I thank you. This document is interesting to read about  Down syndrome Down syndrome – Basque Alzheimer Initiative: clinic-biological cohort

Dear author,

Comments

1.    Introduction: Please specify the aim of the study at the end of the introduction.

2.1 Recruitment

Authors mentioned  individuals with DS aged 18 years or older

Why not DS less than 18 years? Clarify…

2.2. Data collection: The data collection is carried out using forms specifically created for this purpose in RedCap [24,25], a web platform for creating databases, for both qualitative and quantitative variables, collected either during clinical visits or as a result of interpreting the results of complementary tests.

Authors mentioned in the data collection qualitative and quantitative variables, please specify what are those variables….

Kindly mention the details of data collection form what are the details collected.……….

2.8

Line no 282… authors mentioned 15 mL are extracted for storage at -80ºC, intended for future research purposes.

What is the need for this storage? Which tests are done in the future? Where would provide the results?

Sleep study

Authors mentioned about sleep study Nothing mentioned related to the sleep  study …When it is done, How it is performed, what is the duration?

Mention the parameter to be estimated using sleep study..

On what basis obstructive sleep apnea is diagnosed?

Who will perform sleep study?

Discussion

Clinical implication of research to be discussed more with various published literatures.

Author Response

First and foremost, thank you very much for your suggestions aimed at improving our work. Below is our response to your comments:

  1. At the conclusion of the introduction, we have included a paragraph that outlines the purpose of the study.

  2. We have provided an explanation for the exclusion of individuals under 18 years of age in the text.

  3. Detailed information regarding the variables collected in the RedCap forms has been incorporated, organized into three sections: neurological-neuropsychological visits, tests, and research-teaching interests.

  4. The purpose of storing the surplus sample of cerebrospinal fluid (CSF) has been clarified.

    1. We have provided more detailed specifications regarding the sleep assessment, covering both subjective perception evaluation and objective sleep study, while also explaining the limitations of the current initiative.

    2. The discussion section has been expanded in accordance with your recommendations.

    Please let us know if there are any further adjustments or additions you would like us to make.

Reviewer 3 Report

Comments and Suggestions for Authors

The problem of providing medical and social care to adult patients with Down syndrome is beyond doubt due to the increased life expectancy of people with a genomic mutation underlying the development of this chromosomal pathology. Cognitive disorders and decreased intelligence of varying severity are among the well-known clinical manifestations of Down syndrome. The addition of Alzheimer's type cognitive disorders can contribute to a deterioration in the quality of life of patients themselves and their relatives (or caregivers). However, the incidence of comorbid Alzheimer's disease in adult patients with Down syndrome needs to be clarified. Nevertheless, the Basque Alzheimer Initiative seems to me important and timely.

The manuscript needs revision and technical correction.

Abstract: Please focus your attention on the Basque Alzheimer Initiative and its features, specific results, in order to increase visibility and reader interest in the article. In its present form, the abstract contains mostly general information about the problem.

Introduction: Add the results of epidemiological studies demonstrating the incidence of Alzheimer's disease in patients with Down syndrome. Does the nature of the causal genomic mutation underlying the development of Down syndrome affect the frequency and severity of Alzheimer's type comorbid cognitive disorders? In which countries have epidemiological studies been conducted? Formulate and add the purpose of your research.

Materials and Methods: Add this title for section 2. In subsection 2.2. add the characteristics of the study participants, the criteria for inclusion and exclusion. Remove abbreviations from the names of sections and subsections. Instead of "neuropsychological visit", it is more correct to write " neuropsychological assessment" (subsection 2.5). Add a note to Table 2 and explain all the abbreviations you use. What diagnostic equipment was used for video EEG monitoring? What diagnostic equipment was used to diagnose obstructive sleep apnea?

Results. Add this name to section 3. Add a table with characteristics of participants with Down syndrome, including minimum and maximum age depending on gender, average age depending on gender, the nature of the causal genomic mutation, average neuropsychological testing (according to the scales and questionnaires you used), the severity of sleep apnea/hypopnea syndrome, and so on. The differences in the obtained indicators in patients with Down syndrome with and without Alzheimer's type cognitive disorders are of interest. In its present form, the Results section is the "weak link" of the article.

Comments on the Quality of English Language

The English style needs a moderate correction.

Author Response

First and foremost, thank you very much for your thorough review of our work and for the suggestions made to enhance its quality. Here is our response to your comments:

  1. The abstract has been restructured.

  2. Additional information has been provided regarding other initiatives that have inspired DS-BAI, along with data on the prevalence of symptomatic DSAD and age of symptomatic onset described in other cohorts.

  3. The objective of the work has been specified. Our aim is not to provide a comprehensive characterization of the recruited sample since recruitment is ongoing, and follow-up is uneven for all participants at present. The real objective is to present the protocol of the initiative and provide only preliminary data related to the implementation of the initiative to confirm its feasibility. Therefore, while we have provided more data on the characteristics of the sample, our intention is not to conduct a comprehensive characterization of neuropsychological evaluation, findings in structural neuroimaging, and biochemical biomarkers at the current stage. Consequently, we do not believe that Section 2 should be labeled "Materials and Methods" and Section 3 "Results," as the design varies in this case since it involves presenting a protocol and supporting data for its feasibility.

  4. More detailed information on sleep analysis and video-electroencephalography has been incorporated.

  5. The limitations of the present study have been better specified, and more emphasis has been placed on the potential research utility of this initiative.

  6. Information on the abbreviations present in Table 2 has been added.

Please let us know if there are any further adjustments or additions you would like us to make.

Round 2

Reviewer 2 Report

Comments and Suggestions for Authors

Authors fulfilled the comments.

Author Response

Thank you for your kind review.

Reviewer 3 Report

Comments and Suggestions for Authors

The authors have modified the manuscript, but it still needs a technical revision:

1) add a note under Table 1 and explain all the abbreviations you used;

2) move the note explaining the abbreviations under Table 2 (below this table);

3) lines 291, 324, 359 - remove the abbreviation from the name of the subsection, remove the colon.

Author Response

Thank you very much for your constructive comments. We have implemented all the suggested changes, marked with track changes in the document. Consequently, the information on all abbreviations used in the tables is now provided at the bottom of each table, and the ":" symbol and the abbreviations in the titles of the subsections have been removed.